# Study on Micro Interfacial Charge Motion of Polyethylene Nanocomposite Based on Electrostatic Force Microscope

**DOI:** 10.3390/polym11122035

**Published:** 2019-12-09

**Authors:** Bai Han, Jiaxin Chang, Wei Song, Zhi Sun, Chuqi Yin, Penghao Lv, Xuan Wang

**Affiliations:** 1Key Laboratory of Engineering Dielectrics and Its Application, Ministry of Education, Harbin University of Science and Technology, Harbin 150080, Heilongjiang, China; songwei791214@163.com (W.S.); sunzhimems@163.com (Z.S.); ChuqiYinhust@163.com (C.Y.); lvpenghao95@163.com (P.L.); wangxuan@hrbust.edu.cn (X.W.); 2State Key Laboratory Breeding Base of Dielectrics Engineering, Harbin University of Science and Technology, Harbin 150080, Heilongjiang, China; 3College of Electrical & Electronic Engineer, Harbin University of Science and Technology, Harbin 150080, Heilongjiang, China

**Keywords:** electrostatic force microscope (EFM), interface area, space charge, SiO_2_/LDPE nanocomposite

## Abstract

The interface area of nano-dielectric is generally considered to play an important role in improving dielectric properties, especially in suppressing space charge. In order to study the role of interface area on a microscopic scale, the natural charge and injected charge movement and diffusion on the surface of pure LDPE and SiO_2_/LDPE nanocomposite were observed and studied by gradual discharge under electrostatic force microscope (EFM). It was detected that the charge in SiO_2_/LDPE nanocomposite moved towards the interface area and was captured, which indicates that the charge was trapped in the interface area and formed a barrier to the further injection of charge and improving the dielectric performance as a result. Moreover, pulsed electro-acoustic (PEA) short-circuited test after charge injection was carried out, and the change of total charge was calculated. The trend of charge decay in the EFM test is also generally consistent with that in PEA short-circuit test and can be used to verify one another. The results revealed the law of charge movement and verified the interface area can inhibit space charge injection in nano-dielectric at the microscale, which provides an experimental reference for relevant theoretical research.

## 1. Introduction

High voltage direct current (HVDC) transmission system has the advantages of large transmission capacity, small loss, good stability, and low cost. The research and development of HVDC cables have become very important with the urgent demand for urban power transmission cables with larger capacity and higher voltage. In this development process, the most critical problem is the insulation strength of polyethylene material and the space charge accumulation under a high electric field [1]. Electric charges are easily injected into the middle of the insulating material under the condition of high electric field, resulting in the accumulation of space charge [2]. The existence of a large amount of space charge will form the distortion of the local electric field, which will lead to such problems as decreasing breakdown voltage, accelerating the growth of electrical tree, water tree, and aging. In order to solve the problem of space charge accumulation of insulating materials under a high electric field, the research on nano-dielectric has developed rapidly [3,4,5]. Many studies have shown that doping with nanoparticles is beneficial to improving the charge distribution of polymers in a high electric field. For example, nano SiO_2_, MgO, ZnO, and other inorganic oxide doping can effectively improve the dielectric properties of composite materials [1,5,6]. However, the research on the mechanism of improving dielectric properties of nano-dielectric has not been universally accepted so far. Representative theories are as follows, in order to explain the nano-dielectrics with the excellent electrical properties by doping nanoparticles, Lewis [7] proposed a two-layer model in nano-dielectrics to recognize and exploit the unique qualities of nano-dielectric. Tanaka [8] proposed a multi-core model based on the colloid theory to describe the multifaceted characteristics of polyamide nanocomposites. These theories cannot perfectly explain the various phenomena of nano-dielectric and have no universality. Nevertheless, the importance of the interface area between nanoparticles and polymer matrix is emphasized in both of them. In subsequent studies, most of the researchers believed that the improvement of space charge injection of nano-dielectrics under high electric field mainly depended on the action of interface area formed between nanoparticles and polymer matrix [9,10]. The interface area can provide more traps and trap level in nano-dielectrics to capture charge, which cannot be moved easily under high electric field, which will form a blocking electric field and reject the charge further injection, and prevent the distortion of the local electric field [11]. Although this conclusion is generally accepted by many studies, there is a lack of research on the microscopic physical phenomena of the interface, especially the charge injection distribution and movement around the interface area. This is mainly due to the difficulty in observing the structure and charge distribution at the micro-scale of the interface area, especially the law of charge motion is difficult to grasp. 

Scanning probe microscope (SPM) is a kind of high-resolution microscope with multiple scanning modes, and its electric force microscope (EFM) mode is an effective method to observe the distribution of electric field (charge) at micro-scale [12]. By using various scanning modes of SPM, including EFM mode, some researchers studied various characteristics and rules of dielectric parameters of nanoparticles in nano-dielectric, including local dielectric spectroscopy, charging response and polarization properties. Labardi used EFM to detect the local dielectric constant around nanoparticles in nano dielectric [13]. Borgani applied charge and discharge processing in LDPE/Al_2_O_3_ and characterized the trap energy level on Al_2_O_3_ nanoparticles by EFM [14]. Peng used EFM to measure the dielectric constant of the interface area around nano-TiO_2_ in LDPE/TiO_2_ [15] and the polarization around nanoparticles in P(VDF-TrFE)/BT [16]. These results have provided important experimental data for interface research in nano-dielectric. However, most of these studies are carried out under the condition of steady state of charge in nano-dielectrics and little attention is paid to the movement of charge around nanoparticles in nano-dielectrics.

In the study of nano-dielectric, the charge distribution on the micro-scale and the change rule with time can reveal and characterize the charge distribution characteristics in nano-dielectric, especially near the interface area observed by EFM, which provides strong support for the theory of interface area. In this paper, the charge distribution and motion characteristics in LDPE and SiO_2_/LDPE nanocomposite are studied under suitable discharge treatment and observation time. In this paper, LDPE is chosen as a matrix material because it is a representative insulating material in HVDC cables and its structure is relatively simple [14]. The natural discharge process and the charge distribution and change process after charge injection of LDPE and SiO_2_/LDPE nanocomposite especially the change of charge distribution around nanoparticles were studied continuously under EFM observation. The change law of the charge in the nano-dielectric including the charge around the nanoparticle can reflect the trap characteristics inside the material, so as to verify the important role of the interface area after the nano-SiO_2_ doping. Moreover, to verify the EFM experiment results, pulsed electro-acoustic (PEA) method under short-circuited discharge is used to determine whether this charge movement law occurs locally or globally in the nanocomposite. The consistency of EFM and PEA results can prove the reliability of EFM micro-area experimental results. 

## 2. Experimental

### 2.1. Materials Preparation

LDPE produced by Daqing petrochemical company (Daqing, China) and model M-5 gas phase nano-SiO_2_ produced by Cabot company (Boston, Massachusetts, US) were used in this experiment. The diameter of nano-SiO_2_ is 100 nm. The LDPE was melted with Hapro RM-A200 rheometer (Harbin, China) at 130 °C, and then the dispersion of SiO_2_/alcohol was then dripped into it. The doping content of SiO_2_ is 0.5 wt %. The rotor speed was set to 60 r/min and the material was continuously stirred until the torque of the rheometer is constant to evenly disperse the SiO_2_ in LDPE. A flat vulcanizing machine was adopted to form the samples into films. In particular, the EFM (Bruker Multimode 8, Billerica, MA, USA) test sample was formed at a thickness of 50 μm with a mica film with a super flat surface. The advantage of using mica is using its super flat surface to make the surface of samples as smooth as possible. It can expose the part of the doped nanoparticles and help to eliminate the influence of surface fluctuation on charge injection and charge measurement. The PEA test samples were coated with 25 mm diameter aluminum electrodes on both sides and placed in a 50 °C oven for 24 h under short-circuit to eliminate the internal charge.

### 2.2. EFM Measurement

EFM is based on the tapping mode and lift mode of SPM [17], its working principle is shown in Figure 1. First, tapping mode was used to scan the surface morphology of the sample, as shown in Figure 1a. In order to prevent the sample surface from being damaged by the tip, the probe vibrated on the sample surface and touched the sample surface lightly during the scanning. Next, the reflected signal of the laser applied on the probe collected by a photodetector was recorded to detect the displacement of the probe, and then, reflect the surface morphology of the sample. After tapping mode was used to scan a line of the sample surface, lift mode was used to scan the same line again, as shown in Figure 1b. In lift mode, the probe was not in contact with the sample surface but was driven to vibrate at a fixed height (lift height) above the sample surface [18]. At this point, the probe followed stored surface morphology at the lift height above the sample while responding to electric influences between the tip and sample surface on the second scan. A photodetector was also used to collect the signal and record the motion of the probe, as well as record and analyze the phase of the probe vibration waveform to present it on the phase diagram. Therefore, the phase diagram can reflect the electric field force distribution. 

After the samples were pressed into films, there was a large amount of surface charges on the surfaces of pure LDPE and SiO_2_/LDPE nanocomposite, which may have been generated during high-temperature processing, or generated by the friction charge between mica and the surface of the samples [19]. These surface charges remained on the surface and were difficult to dissipate, which exerted a strong electric field force on the probe during the EFM test, making it impossible to test surface charge. This was also the difficulty of surface electric field force test for LDPE all along. 

In order to observe the discharge process of the natural charge on the surface of the sample, the method of gradual discharge of surface charge was used to reduce the influence of a large amount of surface charge in the test. The process was to stick the back of the sample on the metal sample table by silver conductive tape making the side of the EFM test to be measured facing up. Then, a clean flat electrode was placed on the sample surface and connected with the metal sample table by wire and grounded at the same time to release the surface charge. The sample was discharged at 50 °C until 24 h, the phase of the surface could be detected. Then, the surface of the sample was scanned by EFM mode, and the morphology and phase diagram were obtained. In the EFM testing process, in order to avoid the influence of van der Waals force and to better detect the change of electrostatic force, the lift height of the interleave was set as 20 nm [20]. After scanning, the sample was discharged again as mentioned above until discharge time reached 48 and 72 h. Under the condition of retaining the original parameters, the in-situ area of the sample was found and scanned again in EFM mode. By using the software of SPM (Bruker Multimode 8, Billerica, MA, USA), phase data of the same line on the sample were extracted after different short-circuit times. 

After the experiments mentioned above, the SPM probe was used to inject charge into the surface of the sample, and then the change of surface charge was observed by EFM. The experimental processes were as follows: First, the surface of the sample was cleaned with pure water, and the surface charge was fully released without damaging the surface morphology of the sample. Then, the sample was placed in an oven at 50 °C to dry for 1 h, and the sample was stuck on the surface of the sample table with a conductive tape. SPM contact mode was then adopted to contact the sample with the tip and inject charge. The tip bias of −12 V and injection range of 500 nm × 500 nm were adopted. Finally, EFM mode was used to perform in-situ characterization of the sample surface with the same parameters as before and scanning was repeated for 3 h continuously to observe the movement of charge. The EFM scanning observation range was set to 5 μm × 5 μm and probe lift height was set to 20 nm.

### 2.3. Space Charge Measurement

In order to provide a macroscopic data reference for the charge variation in the sample, the charge variations of the whole samples during short-circuit conditions were measured. The space charge measurement of the specimen was performed in a PEA system. The PEA method is a method of measuring the charge distribution in the direction of material thickness. A high voltage source was used to apply a polarization voltage to the upper side of the sample and a pulse source was applied to the sample to generate electrical pulses regularly. According to the pulsed electro-mechanical stress effect, when the electrical pulse was applied, the charge in the sample generated a pressure wave and conduct to the lower electrode [21]. Since the propagation of the electrical pulse is much faster than the pressure wave, the pressure wave of the whole sample is considered to be generated simultaneously. According to the distance to the lower electrode, the pressure wave at each position propagates to the lower electrode in order. A PVDF piezoelectric sensor was placed under the lower electrode to convert the pressure signal into an electrical signal. By analyzing the process of pressure signal propagation and superposition, the obtained electrical signal waveform was processed to obtain the vertical distribution of charge in the sample. When the voltage was applied for a certain period of time, the high voltage source could be removed so that both ends of the sample were grounded. The space charge distribution at different short-circuit times can be detected. In the PEA test, an electric field with the electric field intensity of 50 kV/mm was applied and kept for 30 min so as to make the charge injected. After that, the sample was short-circuited and the change rule of charge distribution with the increase of time was recorded. Then, the space charge distribution of a short circuit was used to calculate the residual charge inside the sample at 10 s, 15, 30 min, 1, 2, 3, 24, 48, 72 h after a short circuit. Although the PEA method was used to measure the charge distribution and charge amount of the whole sample, the charge near the surface of the sample occupied most of it [5,8]. Meanwhile, the charge distribution and total amount in the sample corresponded to the charge distribution and amount near the sample surface [22]. Therefore, the change rule of residual charge in the sample by PEA test can provide reference and correspondence for surface charge variation in the EFM test.

## 3. Results and Discussion

The information of the EFM phase is very complicated. While the lift height is *z*, and the detected location is unchanged, according to the widely accepted mathematical model [23], the equation can be drawn as Equation (1):(1)$$\Delta \varphi \approx \frac{Q}{k}\frac{\partial F\left(z\right)}{\partial z}=\frac{Q}{k}\frac{q_{s}}{4\pi \varepsilon_{0}z^{3}}\left(q_{t}+2CV\right)$$
where Δφ is the phase, Q is the quality factor of the tip, k is the elasticity modulus of the tip, $q_{s}$ is the amount of charge on the sample surface, $q_{t}$ is the amount of charge on the tip. And $V=V_{c}+V_{tip}$, $V_{c}$ is the work function difference between the tip and the sample surface, $V_{tip}$ is the Tip Bias. When the same area is scanned repeatedly, $V_{c}$ does not change. In other EFM experiments, the influence of $V_{c}$ cannot be completely eliminated. Because $q_{t}$ depends on $q_{s}$, only if $q_{s}$ changes, Δφ will change. It can be inferred that the different measured phase values Δφ in the same region must be caused by the change of $q_{s}$. It should be pointed out that Δφ is not proportional to $q_{s}$, but proportional to $q_{s}^{2}$ [24]. It can be considered that the phase information in EFM mode can be used to represent the information of the local charge amount. 

Surface morphology of pure LDPE is shown in Figure 2a. The phase diagrams of discharge for 24, 48 and 72 h are shown in Figure 2b–d. Phase data at different times were selected from a horizontal line (white line) in the middle of the figure and were shown in Figure 2e. As it is shown in Figure 2a, pure LDPE surface morphology is relatively flat, with no obvious fluctuation. It can be seen from the surface morphology that the structure of polyethylene crystalline lamellae or spherulites (in the light color area) is disorganized in the matrix and there is no obvious rule [25,26]. After discharge for 24 h, the surface charge of LDPE was not significantly concentrated and evenly distributed. After 48 h of discharge, the surface charge of LDPE dissipated obviously, and the residual charge was more difficult to detect. After 72 h of discharge, the surface charge could not be detected, and the phase signals were all noises. It indicates that the surface charge of LDPE had completely dissipated. The charge line distribution rule shown in Figure 2e also fully proves these phenomena in pure LDPE. When the discharge time reaches 24 h, the surface charge distribution curve has some fluctuates, but the amplitude is small. There is no obvious correspondence between the surface charge distribution curve and the morphology. When the discharge time reaches 48 and 72 h, there is only a slight difference between the phase signal and noise, which means there is no obvious charge distribution on the surface.

Similarly, the surface morphology of SiO_2_/LDPE is shown in Figure 3a, and the phase diagrams of discharge for 24, 48, and 72 h are shown in Figure 3b–d. As it is shown in Figure 3a, SiO_2_/LDPE has almost the same surface structure as LDPE, its surface is flat. However, unlike LDPE, an exposed nano-SiO_2_ particle (in red circle) was detected at the lower-left corner of the diagram. It can be observed that the nanoparticle’s diameter is about 150 nm. In order to visually reflect the line distribution and relative amount of charge, a straight line (white line) was selected to cross the nanoparticles and the interface area. Phase data extracted from the line are shown in Figure 3e. After 24 h discharge, charges were concentrated in several areas, including the area around the nanoparticles. Meanwhile, concentrated charges were also detected in the region out of the exposed nanoparticle. It might be the charge at the interface area around the unexposed subsurface nanoparticles [27]. Regarding the concentrated charge marked in a blue circle, each charge bright spot represents a nanoparticle which is not exposed, and the distribution and the size of these bright spots indicate that the nanoparticles are evenly distributed in the matrix. After 48 h discharge, as it is shown in Figure 3c, it could be found that the charge of the subsurface and the exposed nanoparticles decreased. After 72 h, the phase around the subsurface nanoparticle disappeared. However, the phase of nanoparticles in Figure 3e can still be distinguished from the matrix phase, which may be due to the difference in work function between nanoparticles and the LDPE matrix [23]. Compared with the pure LDPE, the phase range of SiO_2_/LDPE in Figure 3e is significantly larger, and its peak value even ten times larger than that of the LDPE.

The above experimental results show that, compared with LDPE, the surface of SiO_2_/LDPE is more likely to accumulate charge, especially in the interface area around the nanoparticles. Meanwhile, the charge dissipation rate of SiO_2_/LDPE is significantly slower than that of LDPE, which indicates that the interface area generated by the doping of nano-SiO_2_ particles is more likely to capture electrons. It can be considered that there are a lot of traps in the interface area induced by nano-SiO_2_ particles doping, which makes it difficult for the trapped electrons to escape, forming the accumulated charge and making it difficult to dissipate [28]. 

After the sample was fully discharged, the charge was injected into the surface of the sample by the method mentioned above, and then, the surface of the sample was continuously scanned by EFM mode. After charge injection, the surface morphology of LDPE is shown in Figure 4a, the trace produced by the contact between the probe and the surface of the sample during the charge injection can be observed. Phase figures of 10, 20, 30, and 180 min after charge injection are shown in Figure 4b–e. The same as the previous method, a straight line was selected through the charge injection region, the data were extracted on it, and Figure 4f was drawn. Ten min after injection, the charge was found in the charge injection region and its surrounding region. When discharge time reached 20 min, the phase of the injection region decayed to 1/3 of that at 10 min. The charge around the injection region decreased significantly, and it spread isotropically. When the discharge time reached 30 min, the phase data of the injected area could not be distinguished from the surrounding area, and almost no charge residue could be seen. Furthermore, after the discharge time reached 180 min, the phase of the charge injection area was completely consistent with the surrounding area. The surface charge of LDPE dissipated very quickly after the injection, and it was difficult to stay on the surface. This trend of charge variation in LDPE could also be clearly observed in Figure 4f. 

Figure 5a is the surface morphology of SiO_2_/LDPE after charge injection. It can be seen from Figure 5a, different from LDPE, there is an exposed nanoparticle on the surface of SiO_2_/LDPE shown in the red circle. The phases of 10, 30, 60, and 180 min after charge injection are shown in Figure 5b–e. A straight line was connected between the nanoparticles and the charge injection point, the charge data were extracted on this line, and a graph at different discharge times was drawn, as shown in Figure 5f. Ten min after charge injection, the maximum phase of SiO_2_/LDPE surface was only about 1/4 of that of LDPE. At the same time, the phase of nanoparticles was slightly higher than that of the matrix but with little difference. After 30 min, the surface phase decayed slightly, while at the same time, the surface charge of LDPE basically decayed to undetectable. Obviously, the surface charge decay rate of SiO_2_/LDPE was much slower. When the discharge time reached 60 min, the phase peak value of the injection region had no significant attenuation in SiO_2_/LDPE. However, in the area around the nanoparticles, charge began to accumulate even more obviously than the injection region, as shown in Figure 5d. Charge accumulation in the non-injected region indicates that the interface area formed around the nanoparticles has a significant charge trapping effect. It is worth pointing out that the interface area around the nanoparticle that accumulates charge is significantly larger than the size of the nanoparticle itself, suggesting that the range of the interface area is much larger than other theories had predicted. Hundred eighty min after charge injection, both the phase in the charge injection area and in the interface area around the nanoparticle were reduced slightly and could still be clearly distinguished from the matrix. It means that, as time goes on, the charge dissipates slowly. At this time, in LDPE, the existence of an injected charge could not be detected at all.

The above phenomena indicate that the total amount of charge injected into SiO_2_/LDPE nanocomposites is smaller than pure LDPE, but the charge dissipates faster in pure LDPE. Moreover, the charge in SiO_2_/LDPE is more likely to move and accumulate in the interface area around SiO_2_ nanoparticles and dissipate more slowly. This suggests that the interfacial region around the SiO_2_ nanoparticles in the composite introduces more traps in the interface area due to the doping of nano-SiO_2_. These traps are more likely to trap free-moving electrons and limit their dissipation. It is the presence of these trapped charges that increases the surface barrier of SiO_2_/LDPE and inhibits the charge injection under high electric field, in this way, dielectric properties of SiO_2_/LDPE nanocomposite improved [29].

In order to obtain the magnitude of the local electric field of the charge injected by SPM contact mode, the physical field simulation of charge injection was conducted by COMSOL Multiphysics and the simulation results are shown in Figure 6. In the simulation, the radius of the tip was set as 30 nm, side angle of the tip was set as 22.5°, the voltage between the probe and sample is set as −12 V. The dielectric constant of the sample is set as 2.3, and the thickness of the sample was 50 μm. These parameters were derived from the EFM test used in the above experiments. Without considering the effect of work function, when the insertion depth of the probe was 18.5 nm, it can be concluded that the maximum electric field intensity of the tip can reach 84.75 kV/mm, the average electric field intensity of 100 nm around the tip is 34.13 kV/mm. This electric field intensity is generally consistent with the electric field intensity of space charge distribution under a high electric field test by means of PEA. The space charge distribution characteristics obtained in the discharge of PEA method can be used to calculate the variation of charge density in a short-circuit. Although there is a certain difference between the two experiments, according to the electric field intensity results obtained from the electric field simulation, the electric field intensity is basically at the same value. Moreover, the conductive probe has been in contact with the sample during the EFM test, which can be considered as similar to the experimental conditions of PEA discharge in microscale. 

The average charge density of samples measured by the PEA method is calculated by Equation (2) [30]: (2)$$Q\left(t;E\right)=\frac{1}{x_{0}-x_{1}}\iint_{x_{0}}^{x_{1}}\left|Q_{p}\left(x;t;E\right)\right|dx$$
where $Q_{p}\left(x;t;E\right)$ is the space charge density when time is t, field strength is E and the positon is x. Because of different polarity, here taking the absolute value. $x_{0}$, $x_{1}$ is the location of the cathode and anode, respectively. t is the short circuit time after the polarization electric field is removed and E is the polarization electric field. The average charge volume density of the sample at different times is shown in Figure 7. It can be found that the internal charge density (total charge) of SiO_2_/LDPE is half of that of LDPE at the first 10 s after a short circuit. With the increasing of short circuit time, the charge and its density in pure LDPE decreased rapidly, while the charge in SiO_2_/LDPE dissipated slower. As the discharge time continually increased, the charge density in SiO_2_/LDPE was gradually higher than that in pure LDPE. In PEA short-circuit test, the doping of SiO_2_ particles in SiO_2_/LDPE nanocomposites introduced a large number of traps and captured electrons under the action of the electric field, so that a large amount of homopolar charges accumulated near the electrode. These charges formed a high potential barrier near the electrode, making it difficult to inject the charge further [31]. Although pure LDPE has a higher initial charge density, its charge dissipation rate is faster than that of SiO_2_/LDPE nanocomposite. After discharge for about 30 min, the amount of residual charge in SiO_2_/LDPE was more than that of pure LDPE. During the discharge lasting for three days, the charge amount in SiO_2_/LDPE was always higher than that in pure LDPE. The phenomenon that the space charge dissipation of SiO_2_/LDPE is slower than that of pure LDPE is basically consistent with the result observed in EFM. Therefore, there is a certain similarity between EFM charge dissipate test and the PEA short circuit test, which can be used for their mutual verification.

## 4. Conclusions

In summary, this paper adopts the method of gradual surface charge discharge and observes the change rule of naturally existing charge dissipation in pure LDPE and SiO_2_/LDPE nanocomposite by EFM. In addition, charge is injected into the same sample and the change process of charge is observed. It is found that no matter the dissipation of natural charge or injected charge, the dissipation process of charge in SiO_2_/LDPE nanocomposite is significantly slower than that of pure LDPE, and charge in SiO_2_/LDPE was mainly concentrated around nanoparticles, that is, gathered near the interface area. Moreover, the short-circuit discharge test through PEA also proves this trend on the macro scale. This proves that the interface area plays an important role on the scale of microscopic observation, indicating that there are a lot of traps in the interface area. The movable charge is trapped in a high electric field and the trapped charge makes it more difficult to inject charge into the dielectric, thus improving the insulation performance of nano-dielectric. The results revealed the law of charge movement and verified the effect of interface area on the inhibition of space charge injection in nano-dielectric at the micro-scale, which provides an experimental basis for relevant theoretical research.

## Figures and Tables

**Figure 1 polymers-11-02035-f001:**
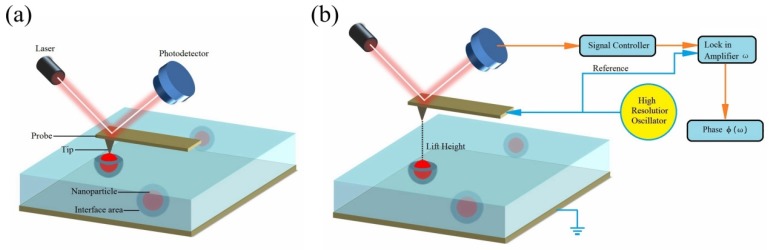
Working principle of electric force microscope (EFM), (**a**) tapping mode in EFM, (**b**) lift mode in EFM.

**Figure 2 polymers-11-02035-f002:**
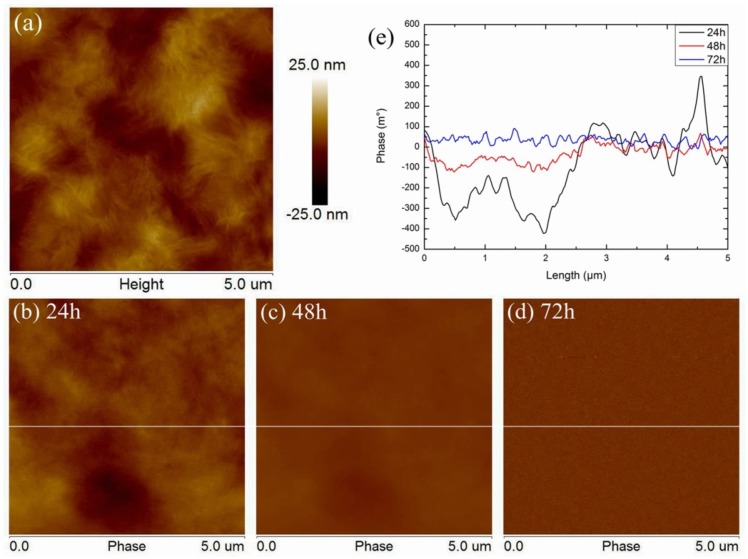
LDPE surface morphology and phase, (**a**) surface morphology, phase after discharge for 24 h (**b**), 48 h (**c**), 72 h (**d**), and phase data (**e**) detected from white line.

**Figure 3 polymers-11-02035-f003:**
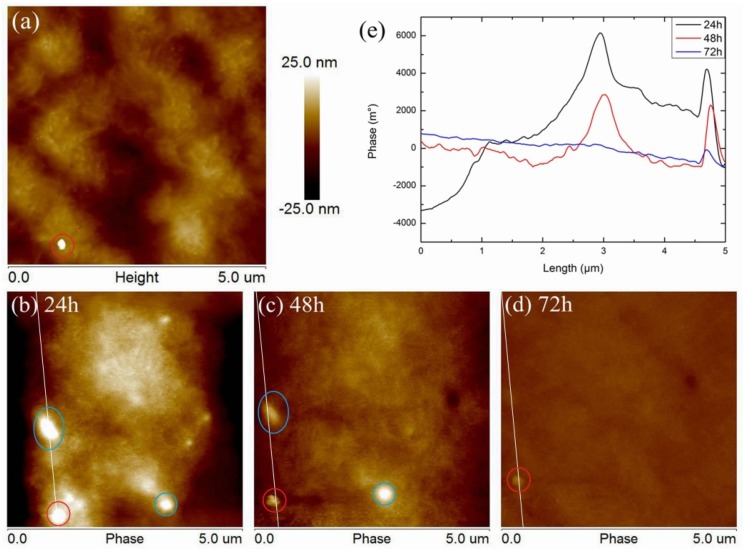
SiO_2_/LDPE nanocomposite surface morphology and phase, (**a**) surface morphology, phase after discharge for 24 h (**b**), 48 h (**c**), 72 h (**d**), and phase data (**e**) detected from white line.

**Figure 4 polymers-11-02035-f004:**
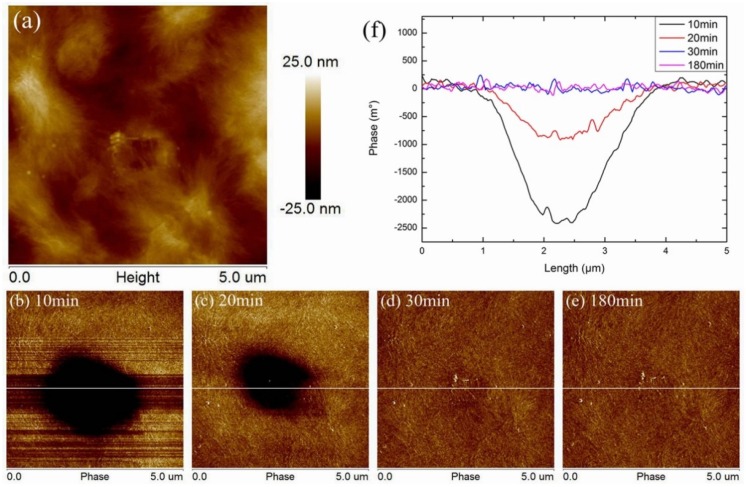
Morphology and phase of LDPE after charge injection, (**a**) surface morphology, phase after charge injection 10 min (**b**), 20 min (**c**), 30 min (**d**), 180 min (**e**), and phase data (**f**) detected from white line.

**Figure 5 polymers-11-02035-f005:**
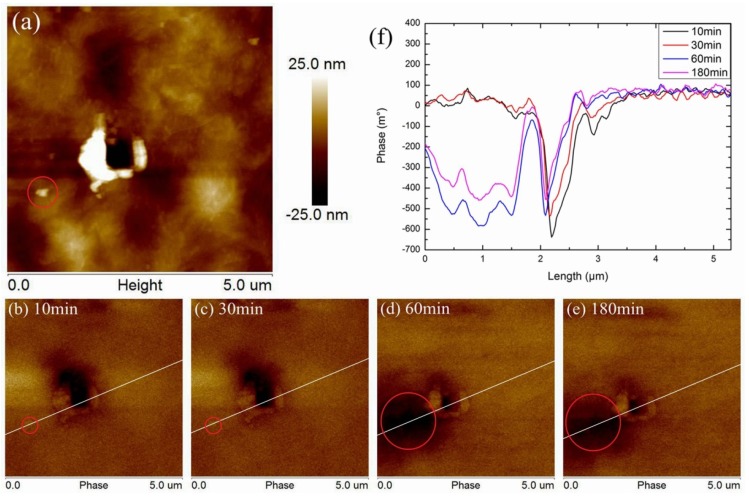
Morphology and phase of SiO_2_/LDPE after charge injection, (a) surface morphology, phase after charge injection 10 min (**b**), 30 min (**c**), 60 min (**d**), 180 min (**e**), and phase data (**f**) detected from white line.

**Figure 6 polymers-11-02035-f006:**
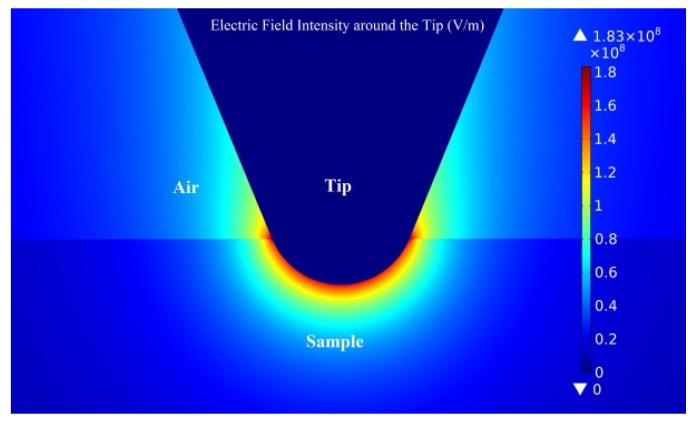
Simulation of the electric field around the tip when charge injecting.

**Figure 7 polymers-11-02035-f007:**
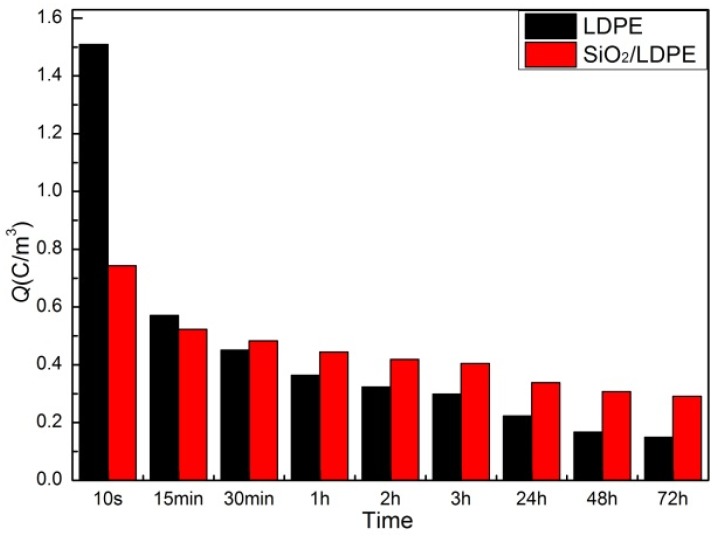
Average charge density of LDPE and SiO_2_/LDPE in short-circuit by PEA method.

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
