# Peer review of "Study on Micro Interfacial Charge Motion of Polyethylene Nanocomposite Based on Electrostatic Force Microscope"

_polymers, 2019, doi:10.3390/polym11122035_

Round 1

Reviewer 1 Report

The authors present a study on charge motion of polyethylene and polyethylene/ SiO2 composites. The study does fall within the scope of the journal. However, while the paper does attempt to present valuable scientific information, it is difficult to comprehend the paper correctly in its current form. Extensive improvements should be done on the language/ sentence structure before it can be considered for publication.

Some specific comments:

Line 24: delete repetitions in the sentence (please do this elsewhere in the paper as well). Line 42: Please replace "of" with "with". Introduction - Please state clearly what is the objective of the current study and what is the main scientific question being addressed. Line 88: Only LDPE will melt while SiO2 would be dispersed in the matirx. Was the distribution of SiO2 homogeneous in the LDPE matrix? On a number of occasions, the SiO2 loaded samples have been referred to as nanocomposites. However, the size of SiO2 particles is 200-300nm. Why are the samples referred to as nanocomposites? Nano scale samples have atleast one dimension less than 100nm. 

Author Response

Dear Reviewer:

Your comments have been taken seriously and the following corrections to your comments were made to manuscript:

Line 23-25: Deleted repetitions in the sentence to make it more concise. Elsewhere in the manuscript, the same changes have been made, including: Line125-128, Line153 and Line381.

Line 36-89: The introduction part had been rewritten to make the background and purpose of the paper more clearly. In the revised introduction section, the main scientific problems of the paper -- the interface problem and its relationship with space charge are described more clearly, and the problem the paper wants to solve -- the law of charge motion at the micro interface is emphasized. At the same time, the errors for "of" and "with" in Line 50 were amended.

Line 90-114: Move the first paragraph of 2. Experimental to the 1. Introduction to make it more clearly.

Line 119-124: In previous manuscripts(Line 88), the mixing process of SiO2 and LDPE was incorrectly described. Corrected it and described the preparation of the material more clearly.

Line 118-119, 248-249: The previous description of nano SiO2 particle size was imprecise. After the verification and measurement of nano SiO2 material parameters, the particle size should be about 100nm. After nano SiO2 doping, the size of the observed nanoparticles will increase appropriately. The size of the nanoparticles shown in the figures of this paper is 150nm. Moreover, considering the scanning resolution of EFM mode, the charge distribution around the nanoparticles of this size can be well characterized. Similar situation exists in other researches’ work:

IEEE Trans. Dielectr. Electr. Insul. 2017, 24, 1027-1037.

Appl. Phys. Lett. 2008, 93, 033109.

IEEE Trans. Dielectr. Electr. Insul. 2014, 21, 004340.

Sci. Rep. 2016, 6, 38978.

Adv. Mater. 2019, 31, 1807722.

The relevant parts of the manuscript are revised.

Line 257-259: Through further analysis of Figure. 3a and Figure. 3b, the dispersion state of nano SiO2 particles is indirectly expressed, and nanoparticles are evenly distributed in the matrix.

By taking your valuable advices, readers will be able to understand the research more clearly.

Reviewer 2 Report

The manuscript entitled "Study on micro interfacial charge motion of polyethylene nanocomposite based on Electrostatic Force Microscope" deals with the characterization of the charge movement and diffusion on the surface of LDPE-based systems through electrostatic force microscope measurements.

In my opinion, the used method is interesting and innovative, the data are clearly described in the manuscript and the conclusions are fully supported by the obtained results. 

Therefore, I suggest the publication of the paper on Polymers.

I recommend to the authors to solve the following minor concerns:

Please specify the meaning of "HVDC" (page 1, line 36); Please, specify the apparatus adopted for the preparation of the composites and add the used processing conditions (Temperature, time of mixing, rpm of the rotors, ecc...)

Author Response

Dear Reviewer:

Your comments have been taken seriously and the following corrections to your comments were made to manuscript:  

Line 36-89: The introduction had been rewritten to make the background and purpose of the research more clearly. At the same time, the meaning “HVDC(High Voltage Direct Current)” was specified(Line36).

Line 90-114: Move the first paragraph of 2. Experimental to the 1. Introduction to make it more clearly.

Line 118-124: The apparatus adopted for the preparation of the composites was specified, and the processing conditions is described in detail.

By taking your valuable advices, readers will be able to understand the research more clearly.

Reviewer 3 Report

In this paper, the authors have observed the charged discharged in pure LDPE and SiO2/LDPE nanocomposite using Electrostatic Force Microscope(EFM). The results shows that the charge in nanocomposites goes towards the interface area and the discharge process is slower than pure LDPE.

Following are some of the suggestions.

Please provide some more information about the basic principal of Electrostatic Force Microscope(EFM) and Pulsed Electro-Acoustic(PEA).

Author Response

Dear Reviewer:

Your comments have been taken seriously and the following corrections to your comments were made to manuscript:

Line 90-114: Move the first paragraph of 2. Experimental to the 1. Introduction to make it more clearly.

Line 134-149: The basic principal of Electrostatic Force Microscope(EFM) is described. Figure 1 is added to illustrate the working principle of EFM.

Line 185-198: The basic principal of Pulsed Electro-Acoustic(PEA) is described.

By taking your valuable advices, readers will be able to understand the research more clearly.

Round 2

Reviewer 1 Report

The manuscript has been revised as per earlier suggestions, hence the manuscript has been improved significantly. Further improvements to sentence structure/ language can be done, however the revisions done should suffice. In my view, the manuscript now can be accepted for publication.